# Pharmacological Inhibition of NHE1 Protein Increases White Matter Resilience and Neurofunctional Recovery after Ischemic Stroke

**DOI:** 10.3390/ijms241713289

**Published:** 2023-08-27

**Authors:** Shamseldin Ayman Hassan Metwally, Satya Siri Paruchuri, Lauren Yu, Okan Capuk, Nicholas Pennock, Dandan Sun, Shanshan Song

**Affiliations:** 1Department of Neurology, University of Pittsburgh, Pittsburgh, PA 15213, USA; smetwally@pitt.edu (S.A.H.M.); satyasiri@gmail.com (S.S.P.); yu.lauren@medstudent.pitt.edu (L.Y.); capuk@pitt.edu (O.C.); ndp35@pitt.edu (N.P.); 2Pittsburgh Institute for Neurodegenerative Disorders, University of Pittsburgh, Pittsburgh, PA 15213, USA; 3Veterans Affairs Pittsburgh Health Care System, Pittsburgh, PA 15213, USA

**Keywords:** Cariporide, inflammation, microglia, myelination, Rimeporide, white matter

## Abstract

To date, recanalization interventions are the only available treatments for ischemic stroke patients; however, there are no effective therapies for reducing stroke-induced neuroinflammation. We recently reported that H^+^ extrusion protein Na^+^/H^+^ exchanger-1 (NHE1) plays an important role in stroke-induced inflammation and white matter injury. In this study, we tested the efficacy of two potent NHE1 inhibitors, HOE642 and Rimeporide, with a delayed administration regimen starting at 24 h post-stroke in adult C57BL/6J mice. Post-stroke HOE642 and Rimeporide treatments accelerated motor and cognitive function recovery without affecting the initial ischemic infarct, neuronal damage, or reactive astrogliosis. However, the delayed administration of NHE1 blockers after ischemic stroke significantly reduced microglial inflammatory activation while enhanced oligodendrogenesis and white matter myelination, with an increased proliferation and decreased apoptosis of the oligodendrocytes. Our findings suggest that NHE1 protein plays an important role in microglia-mediated inflammation and white matter damage. The pharmacological blockade of NHE1 protein activity reduced microglia inflammatory responses and enhanced oligodendrogenesis and white matter repair, leading to motor and cognitive function recovery after stroke. Our study reveals the potential of targeting NHE1 protein as a therapeutic strategy for ischemic stroke therapy.

## 1. Introduction

Stroke is among the most prevalent diseases worldwide; however, despite continuous efforts, many agents that have been tested to be effective in preclinical models have failed in clinical trials due to translation efficacy and safety issues [1]. To date, recanalization interventions to remove the clot with tissue plasminogen activator (tPA) or mechanical endovascular thrombectomy (EVT) are the only treatments available for ischemic stroke patients. Restoring blood flow after prolonged ischemia causes reperfusion injury, which generates reactive oxygen species (ROS) to damage brain cells [2] and exert deleterious effects with a hemorrhagic transformation [3]. Therefore, the current guidelines for recanalization interventions are recommended within 4.5–6 h of the stroke onset [4], although recently the window of EVT has been extended to 6–24 h from the onset of ischemic stroke for patients with a mismatch between the clinical deficit and infarct [5]. The short therapeutic window severely limits stroke patients to receive effective treatments, where only 11.8% receive tPA, and 5.7% receive EVT [6]. Thus, there is an unmet need to develop more effective stroke therapies.

Inflammatory responses play a significant role in brain injury and post-injury recovery following various pathological conditions [7,8,9] including ischemic stroke [10,11]. Currently, no therapy is available for reducing inflammation after stroke [12]. Upon ischemic stroke, microglial cells mediate the main innate immune responses, with morphological changes (such as process retraction, etc.) in the ischemic penumbra from as early as 30 min to 1 h [10] and with increased cell counts peaking at 3–7 d post-stroke [13], which remained elevated at 2–3 weeks post-stroke [14]. During microglial activation, they can release inflammatory cytokines (IL-1β, TNF-α, IFN-γ, iNOS, etc.) [15,16,17] or simultaneously can dynamically regulate their adaptive function by releasing restorative cytokines/growth factors (TGF-beta, IL-10, BDNF, GDNF), clearing ischemic tissue debris through phagocytosis and promoting tissue repair [17,18,19]. The dynamic change in microglial functions is crucial for post-stroke brain repair and functional outcome improvement [13,20,21]. Thus, targeting inflammation including modulating microglial functions presents a promising treatment for stroke with a wider therapeutic window [2].

Na^+^/H^+^ exchanger isoform-1 (NHE1) mediates the electroneutral transport of H^+^ efflux in exchange of Na^+^ influx and thus regulates the intracellular pH (pH_i_), which is essential in the sustained activation of NADPH oxidase (NOX) for ROS productions in post-ischemic neurons and astrocytes [22,23]. We recently reported that NHE1 protein is also required for microglial activation during the respiratory burst [24,25]. Selective deletion of microglial *Nhe1* in *Cx3cr1-Cre^ER+/−^*; *Nhe1^flox/flox^* mice reduced microglial pro-inflammatory responses, elevated their phagocytic activity, enhanced synaptic pruning, and improved white matter myelination, altogether contributing to significant neurological function recovery after ischemic stroke [13,26]. In the current study, we investigated the therapeutic efficacy of the post-stroke administration of two potent NHE1 protein inhibitors Cariporide (HOE642) and Rimeporide, with a delayed administration regimen at 24 h post-stroke. Rimeporide emerges as a first-in-class NHE-1 inhibitor currently undergoing phase I clinical trials for Duchenne muscular dystrophy [27]. We found that post-stroke administration of HOE642 or Rimeporide in adult C57BL/6J mice did not reduce the ischemic infarct volume but reduced microglial inflammation, improved oligodendrogenesis and white matter myelination, and accelerated sensorimotor and cognitive recovery after ischemic stroke. Our study identifies targeting NHE1 protein as a novel strategy to reduce neuroinflammation and white matter damage and improve post-stroke recovery. 

## 2. Results

### 2.1. Administration of NHE1 Inhibitor HOE642 at 24 h Post-Stroke Improved Motor-Sensory and Cognitive Functions in C57/BL6J Mice

We first assessed the therapeutic efficacy of the post-stroke administration of the potent NHE1 inhibitor HOE642 on motor-sensory and cognitive function recovery in C57/BL6 mice. To explore the clinical efficacy for expanding the treatment window for stroke patients beyond the 6–24 h suggested in the DAWN trial [5], we tested a delayed regiment of the administration of a vehicle (Veh, 2.5% DMSO) or HOE642 (0.3 mg/kg/day at 0.025 mg/mL, i.p., b.i.d.) starting at 24 h post-tMCAO (Figure 1A) with an initial dose at 24 h post-stroke, followed by b.i.d. treatment for 7 consecutive days, as we recently reported that HOE642 administration (at the 0.3 mg/kg dose) reduced microglial activation [28] and was neuroprotective in a traumatic brain injury (TBI) mouse model [29]. The HOE642-treated mice did not show a statistically improved survival rate or body weight changes, compared to the Veh-treated mice (Appendix A). Neurological scoring is a well-established neurological-deficit grading system in evaluating neurological deficits after tMCAO, ranging from 0 for no neurological deficit to 8 where stroke-related death occurs [13,30]. In the initial 4 days after ischemic stroke, both Veh- and HOE642-treated mice exhibited similarly poor neurological scoring (*p* > 0.05). However, the HOE642-treated mice started to recover significantly faster from day 5 post-stroke (*p* < 0.05, Figure 1B). These HOE642-treated mice also exhibited significant improvements in motor-sensory functions, with a longer latency of staying on the rotating rods in the rotarod accelerating test as early as 3 d post-stroke (*p* < 0.05), with less missed steps in the foot-fault test at 2 d post-stroke (*p* < 0.05), and with an increased sensitivity to the sticker in the adhesive contact test and faster removal of the adhesive tape in the adhesive removal test at 1 d post-stroke (*p* < 0.01) (Figure 1C). Moreover, at 28 d post-stroke, the HOE-treated mice displayed a significantly improved short-term spatial working memory in the y-maze spontaneous alternation test (*p* < 0.01, Figure 1D) and improved long-term recognition memory in the novel object recognition test (*p* < 0.05, Figure 1E), without affecting locomotor activities or anxiety (Appendix A). These findings suggest that the blockade of NHE1 protein activity by HOE642 starting at 24 h post-stroke significantly improved both motor-sensory and cognitive function recovery after ischemic stroke. 

### 2.2. Post-Stroke Administration of HOE642 Did Not Reduce Ischemic Infarct or Neuronal Death but Enhanced White Matter Myelination 

Given NHE1 protein is ubiquitously expressed in all brain cell types, including neurons, astrocytes, microglia, and OLs [31], we first assessed if the Veh- or HOE642-treated mice displayed differential acute neuronal damage at 3 d post-stroke. TTC staining in Figure 2A,B showed that the Veh- and HOE642-treated mice exhibited a comparable ischemic infarct size and hemispheric swelling (*p* > 0.05). Since neurogenesis in peri-infarct tissues can contribute to better neurological function in stroke mice [13], we further examined neuronal counts in the ischemic core, peri-infarct area in the ipsilateral (IL) hemisphere and the contralateral (CL) hemisphere with immunostaining analysis of dendritic marker MAP2 and neuronal marker NeuN expression. Interestingly, compared to the CL hemispheres, both the perilesion cortex and ischemic core of the Veh- or HOE642-treated mice showed a similar loss in MAP2 intensity or NeuN^+^ neuron counts (*p* < 0.05, CL vs. perilesion or core; *p* > 0.05, Veh vs. HOE) (Figure 2C,D). These findings suggest that the post-stroke administration of HOE642 did not reduce neuron degeneration nor acute infarct formation in the ischemic brains. As no significant difference was detected in gray matter damage, we further investigated whether an improved white matter repair could account for the accelerated neurological function recovery in the HOE642-treated stroke mice. Figure 3A shows that at 3 d post-stroke, the HOE642-treated mice exhibited a 39.3% increase in corpus callosum (CC) thickness (*p* < 0.01, Figure 3A), with higher MBP protein expressions (*p* < 0.01) and APC^+^ mature OL counts (*p* < 0.01) in both the CL and IL hemispheres of the CC compared to the Veh-treated stroke mice (Figure 3B). In addition, the HOE642-treated mice also showed an increased OL genesis with higher NG2^+^Olig2^+^ OPC counts (*p* < 0.0001), elevated Ki67^+^Olig2^+^ OL proliferation (*p* < 0.0001), and reduced Caspase3^+^Olig2^+^ OL apoptosis (*p* < 0.0001) in both hemispheres of the CC at 3 d post-stroke, which persisted to 7 d post-stroke (Figure 3C). These data indicate that a delayed regimen of the pharmacological inhibition of NHE1 protein activity with HOE642 from 24 h post-stroke attenuated OL apoptosis and stimulated OL genesis, leading to improved white matter myelination in post-stroke brains. 

### 2.3. Efficacy of the Novel NHE1 Inhibitor Rimeporide in Improving Neurological Functions in C57/BL6 Mice after Ischemic Stroke

Next, we also explored the efficacy of the novel NHE1 inhibitor Rimeporide (RIM) for ischemic stroke treatment in mice. Due to its short half-life averaging about 3–4 h [27], we utilized a continuous delivery approach of an osmotic minipump to maintain stable plasma levels above pharmacological levels. Osmotic minipumps with Veh (5% DMSO in PBS) or RIM at 2.1 mg/kg (equivalent to 0.3 mg/kg/day) were implanted subcutaneously in mice at 1–7 d after stroke (Figure 4A). The survival rate and body weight changes were similar between the Veh- and RIM-treated mice (Appendix A). An ischemic infarct assessment at 3 d post-stroke showed that RIM treatments did not reduce infarct volume but decreased edema formation (*p* = 0.10) compared to the Veh treatment (Figure 4B,C). In addition, the RIM-treated mice exhibited significantly faster motor function improvements in neurological scoring and the rotarod accelerating test from 2 to 14 d post-stroke (Figure 4D,E) but not in the foot-fault or adhesive tests (Appendix A). These RIM-treated mice also showed significantly improved memory function in the y-maze spontaneous alternation test and the novel spatial recognition test at 30 d post-stroke compared to the Veh-treated mice (*p* < 0.05, Figure 4G–H) and a slight improvement in locomotor activity and reduced anxiety (although this did not reach a statistical significance, Figure 4F). These data suggest that the delayed administration of RIM at 24 h post-stroke is effective in improving neurological functions after stroke. 

### 2.4. Post-Stroke Administration of Rimeporide Enhances White Matter Myelination with Increased Oligodendrogenesis and Reduced Apoptosis

To determine whether the RIM-treated mice exhibited similar effects as the HOE-treated mice, we also assessed white matter changes in the RIM-treated mice at 3 d post-stroke. Figure 5A showed that the RIM-treated mice showed a 1.4-fold increase in MBP protein expression (*p* < 0.01) and a 2.9-fold increase in the APC^+^ mature OL counts (*p* < 0.01) in both hemispheres of the CC compared to the Veh-treated mice (Figure 5A). Concurrently, the RIM-treated mice also exhibited an increased OL genesis with higher NG2^+^Olig2^+^ OPC counts (*p* < 0.01), reduced Caspase3^+^Olig2^+^ OL apoptosis (*p* < 0.0001), and enhanced H3K9me3^+^Olig2^+^ OL differentiation (*p* < 0.001) in both hemispheres of the CC at 3 d post-stroke (Figure 5B). Taken together, these data clearly demonstrate that the delayed administration of Rimeporide to block NHE1 protein activity from 24 h post-stroke stimulated OL genesis and differentiation, while reducing OL apoptosis, leading to improved white matter myelination in post-stroke brains, similar to the HOE-mediated effects.

### 2.5. Reduced Microglia-Mediated Inflammation in Stroke Brains Treated with HOE642 or Rimeporide 

We further evaluated neuroinflammation profiles by investigating changes in IBA1^+^ microglia and GFAP^+^ reactive astrocytes in Veh-, HOE642-, or RIM-treated brains at 3 d post-stroke. As shown in Figure 6A, the Veh-treated brains displayed abundant IBA1^+^ microglia and GFAP^+^ reactive astrocytes in the perilesion cortex. In contrast, the HOE642-treated brains exhibited significantly less accumulation of IBA1^+^ microglia (*p* < 0.01), which showed a quiescent ramified morphology, compared to the amoeboid morphology with retracted processes which represents an activated phagocytic phenotype [32] in the Veh-treated brains. In comparison, the cell counts of GFAP^+^ reactive astrocytes in the perilesion cortex were comparable between Veh- and HOE642-treated brains (*p* > 0.05), though those from the HOE-treated brains displayed thinner processes indicating less astrogliosis reactivity [33] (Figure 6A). We further analyzed the inflammatory profiles of CD11b^+^CD45^+^ microglia/macrophages via flow cytometry. Both Veh- and HOE-treated brains showed similar counts of CD11b^+^CD45^+^ microglia/macrophages; however, those in the ischemic hemisphere of HOE-treated brains displayed significantly increased anti-inflammatory marker Ym1 compared to those from the Veh-treated brains (*p* < 0.05, Figure 6B,C), but not the pro-inflammatory markers (Figure 6C and Appendix A). Taken together, these data demonstrate that the post-stroke administration of HOE642 selectively increased microglial/macrophage anti-inflammatory activation but did not significantly affect astrogliosis in the ischemic brains. 

As for the RIM-treated brains, we detected a similarly mitigated inflammation with less IBA1^+^ microglial cells in the perilesion cortex compared to the Veh-treated brains at 3 d post-stroke (Figure 6D). GFAP^+^ reactive astrocyte counts remain similar between the Veh- and RIM-treated brains (*p* > 0.05). Flow cytometry analysis revealed that stroke induced an elevation in CD11b^+^CD45^+^ microglia/macrophage counts in the Veh-treated brains. However, the RIM-treated brains exhibited significantly reduced CD11b^+^CD45^+^ microglia/macrophage counts, without an apparent change in the inflammatory marker profiling (Figure 6E and Appendix A). These data suggest that the post-stroke administration of Rimeporide also altered the inflammation in stroke brains. 

## 3. Discussion

### 3.1. Expanded Time Window and Therapeutic Efficacy of the Pharmacological Inhibition of NHE1 Protein after Ischemic Stroke

The recent DAWN (DWI or CTP Assessment with Clinical Mismatch in the Triage of Wake-Up and Late Presenting Strokes Undergoing Neurointervention with Trevo) trial expanded the time window for endovascular thrombectomy treatment to 6–24 h post-ischemic stroke [5]. However, the latest guideline for the recommended time window of thrombolysis using tPA is still within 4.5 h of the ischemic stroke onset [34]. In this study, we tested the efficacy of the pharmacological inhibition of NHE1 protein with a wider therapeutic window at 24 h post-stroke in adult C57BL/6J mice. Compared to Veh-control stroke mice, HOE642-treated mice (with a low dose at 0.15 mg/kg, i.p., b.i.d. for 1–7 d post-stroke) showed a significantly improved motor-sensory and cognitive functional recovery through 1–28 d post-stroke, with no adverse effects. HOE642 has been tested in the GUARDIAN (Guard During Ischemia Against Necrosis) phase II/III clinical trials for myocardial infarction, showing a 25% reduction in risk in high-risk patients, and reported no serious adverse events in the Cariporide-treated group (with a dose of 120 mg, intravenous t.i.d. for 2–7 days) over the placebo [35]. However, despite reduced myocardial infarction, it failed in the EXPEDITION (Sodium-Proton Exchange Inhibition to Prevent Coronary Events in Acute Cardiac Conditions) phase III clinical trial due to increased embolic stroke incidents [36], which is likely due to the excessive high dose (180 mg loading dose, then 40 mg/h in 24 h, and 20 mg/h in 48 h, intravenous) and its paradoxical effects on platelet activation [36,37]. Our study with a low dose of HOE642 administered at 24 h post-stroke indicated the therapeutic potential of the pharmacological blockade of NHE1 protein with minimized potential adverse effects. Additional studies are warranted on whether the delayed administration regimen with low doses of NHE1 inhibitors has any adverse effects on platelet or other coagulation dysregulation.

In terms of Rimeporide (EMD-87580), it has a similar half-life of 3–4 h to HOE642 and has recently been tested as a first-in-class NHE1 inhibitor in a phase Ib clinical trial for Duchenne muscular dystrophy (DMD) [27]. Compared to HOE642, it was considered safe and well-tolerated at the highest dose of 300 mg, t.i.d. (oral administration) with promising outcomes [27]. We chose a low dose regimen of 0.3 mg/kg/day after pilot experiments with 0.3, 0.5, and 1 mg/kg/day. To maintain stable plasma levels above pharmacological concentrations, we utilized a continuous delivery approach with osmotic minipumps. Interestingly, the RIM-treated mice also exhibited a faster motor-sensory function recovery in the acute to subacute phase after stroke and better cognitive function in the chronic phase post-stroke but without an improvement in overall survival or infarct volume, similar to many other studies [38,39,40]. These novel findings are encouraging and present the therapeutic potential of Rimeporide in promoting tissue repair and functional recovery after cerebral ischemic stroke.

### 3.2. NHE1 Blockers Attenuate Neuroinflammation in Ischemic Stroke

NHE1 protein is a housekeeping pH regulatory protein, mediating the electroneutral transport of H^+^ efflux in exchange of Na^+^ influx [22]. We recently discovered the differential roles of NHE1 protein in neurons, microglia, or astrocytes in brain damages after ischemic stroke [13,23]. The selective deletion of neuronal *Nhe1* (in *CamKIIa-Cre^+/−^*; *Nhe1^flox/flox^* mice) or astrocytic *Nhe1* (in *Gfap-Cre^ER+/−^*; *Nhe1^flox/flox^* mice) both showed a significantly reduced ischemic infarct and accelerated neurological functional improvements in a mouse tMCAO model [13,23], while the targeted deletion of microglial *Nhe1* in *Cx3cr1-Cre^ER+/−^*; *Nhe1^flox/flox^* mice did not reduce infarct but improved motor-sensory and cognitive behaviors with mitigated acute inflammation and enhanced long-term myelination up to 1 month after ischemic stroke [13,26]. In either HOE642- or RIM-treated stroke mice, no reduction in infarct volume nor neuroprotective effects were observed. As neuronal NHE1 activation occurs early after ischemic injury (within 24 h) [41], we concluded that the delayed administration of either HOE642 or Rimeporide at 24 h post-stroke exerted no neuroprotective effects. We believe that the improved neurological behavior with the post-stroke administration of the inhibitors was via alleviating microglial inflammatory responses and protecting white matter tissues. Mismatches between the infarct volume and the severity of clinical deficits are often related to prolonged inflammatory responses and compromised white matter integrity in ischemic stroke patients [5,42,43]. The peak of microglial responses occurs at 3–7 days post-stroke [13] and remains elevated until 2–3 weeks post-stroke [14]. Particularly, NHE1 protein expression in microglial cells remains upregulated to at least 7 days post-stroke [25], allowing for an extended treatment window. Our delayed paradigm of low dose HOE642 at 24 h reduced microglial inflammatory profiles, while the delayed administration of Rimeporide reduced microglia/macrophage cell counts without significant alteration in their profiles. These early microglia-mediated inflammation changes were accompanied by the increased resistance of white matter tissues to ischemic demyelination injury, with better preserved mature OLs and myelination, along with the early elevation of OL proliferation and differentiation, as well as significantly reduced OL apoptosis. These findings clearly show that targeting NHE1 protein by post-stroke HOE642 or Rimeporide administration provides a novel strategy to modulate microglial function for stimulating white matter repair and post-stroke functional recovery.

Regarding the cellular mechanisms underlying Rimeporide’s protective effects in myocardial infarction models, Rimeporide demonstrated positive results in reducing local inflammation [38] and oxidative stress with attenuated reactive oxygen species production via NADPH oxidase and ERK1/2/Akt/GSK-3β/eNOS pathways [44,45], with no effects observed in reducing the infarct size in hearts [38,39,40]. In addition, Rimeporide has been shown to increase mitochondria respiratory functions with reduced mitochondrial permeability transition [46] and increased biogenesis [47] in myocardial infarction models, which were associated with a decreased mitochondrial vulnerability to exogenous Ca^2+^ mediated by Na^+^/Ca^2+^ exchanger (NCX) activity coupled with NHE1 stimulation [46,48]. This is in line with our recent report that the genetic deletion of *Nhe1* boosted mitochondrial oxidative phosphorylation capacities in microglia after cerebral stroke [26]. Similar protective effects were also observed in models of cardiomyocyte hypertrophy/hereditary cardiomyopathy, where Rimeporide preserved the left ventricle heart functions and prevented early death [49,50]. These studies provide further evidence for the protective effects of Rimeporide after stroke. However, its specific suppression effects on CD11b^+^/CD45^+^ microglia/macrophages in stroke brains require further investigation. 

The weakness of this study lies in the systemic NHE1 protein blockade with HOE642 or Rimeporide administration. OPCs/OLs have comparably high levels of expressions of the *Nhe1* gene [51], which is a dominant regulator for their intracellular pH (pH_i_) [52,53]. The systemic inhibition of NHE1 protein could directly affect OPCs/OLs and exert an impact on remyelination. We observed the apparent improvements in white matter repair with increased CC thickness, MBP protein expression, and mature OL preservation in the NHE1-inhibitor-treated groups. However, whether the pathological upregulation of NHE1 protein occurs in OPCs/OLs after stroke and whether HOE642 or RIM directly inhibit NHE1 protein in these cells to promote remyelination warrants further investigation in future studies. In light of the phenotypes of the global transgenic knockout of *Nhe1* on the increased excitability of the CNS and epilepsy (especially slow-wave epilepsy) in mice [54], it raises additional concerns about the global administration approach of NHE1 inhibitors. The development of more cell-type-specific targeted approaches and specific post-stroke treatment regimens is warranted to minimize these adverse effects.

## 4. Material and Methods

### 4.1. Animals

All animal experiments and procedures were approved by the University of Pittsburgh Institutional Animal Care and Use Committee and performed in accordance with the National Institutes of Health Guide for the Care and Use of Laboratory Animals and reported in accordance with the Animal Research: Reporting In Vivo Experiments (ARRIVE) guidelines [55]. Animals were provided with food and water ad libitum and maintained in a temperature-controlled environment in a 12/12 h light–dark cycle. All efforts were made to minimize animal suffering and the number of animals used.

### 4.2. Transient Focal Ischemia Model

Transient focal cerebral ischemia was induced by 60 min transient occlusion of the left middle cerebral artery (tMCAO) as described before [13,26]. Please see Appendix A for detailed surgical procedures.

### 4.3. Drug Administration 

C57/BL6J mice (male and female) at 2–3 months old (Jackson Laboratory, Strain# 000664) were used in this study. HOE642 (Cariporide, Sigma-Aldrich, St. Louis, MO, USA) or Rimeporide (EMD-87580, MedChemExpress, Monmouth Junction, NJ, USA) was dissolved at 1 mg/mL in DMSO as stock solution and diluted to 0.025 mg/mL in PBS immediately before injection. In total, 2.5% DMSO in PBS was used as the vehicle control (Veh). Veh, HOE642, or Rimeporide was administered at 0.3 mg/kg body weight/day by intraperitoneal (i.p.) injections, with an initial dose at 24 h post-stroke, followed by twice per day (b.i.d.) injections (at least 8 h apart) for 7 consecutive days. To test a continued administration regimen after stroke, osmotic minipumps (Alzet, model# 2001, Durect corporation, Cupertino, CA, USA) were used to deliver Rimeporide, which was dissolved at 5 mg/mL in DMSO as stock solution and diluted to 0.25 mg/mL in PBS immediately before loading. The osmotic pumps were then implanted subcutaneously to achieve a constant deliver at 0.3 mg/kg/day (at a rate of 1.0 µL/h) from 24 h to 7 d post-stroke. In total, 5% DMSO in PBS was used as Veh control in the osmotic pump study.

### 4.4. Neurological Function Tests

Neurological functional deficits in mice were screened in a blind manner with neurological scoring. Sensorimotor functions were measured with a rotarod accelerating test, foot-fault test, adhesive contact test, and adhesive removal test, and cognitive functions were determined with an open field test, y-maze spontaneous alternation test, and y-maze novel spatial recognition test, all considered reliable for identifying and quantifying neurological functional deficits in rodent models [13,26,56]. Please see Appendix A for detailed methods.

### 4.5. 2,3,5-Triphenyltetrazolium Chloride (TTC) Staining

Mice were euthanized with an overdose of CO_2_ at 3 d post-stroke, and the mouse brains were dissected and cut into 4 coronal slices of 2 mm thickness before staining with 2% 2,3,5-triphenyltetrazolium chloride (TTC, Sigma-Aldrich, St. Louis, MO, USA) at 37 °C for 15 min, as described previously [13]. The measurement of infarct volume and brain swelling was performed as described before [13]. Please see Appendix A for detailed methods.

### 4.6. Flow Cytometry

Flow cytometry was conducted to investigate microglial inflammatory profiles at 3 d post-stroke. Briefly, single-cell suspensions were obtained from contralateral (CL) and ipsilateral (IL) brain tissues using an enzymatic tissue dissociation kit (Miltenyi Biotech Inc., Gaithersburg, MD, USA) before centrifuging through a 30/70 Percoll (GE Healthcare, Chicago, IL, USA) gradient solution to remove myelin, as described before [13,26]. Cells were subsequently stained with BUV395-conjugated CD11b (1:400, BD Biosciences, San Jose, CA, USA), PerCP-Cy5.5-conjugated CD45 (1:400, BioLegend, San Diego, CA, USA), eFluor450-conjugated CD16/32 (1:400, Affymetrix eBioscience, San Diego, CA, USA), FITC-conjugated CD206 (1:400, BioLegend, San Diego, CA, USA), Alexa 700-conjugated CD86 (1:400, BD Biosciences, San Jose, CA, USA), PE-conjugated Ym1 (1:400, Abcam, Boston, MA, USA), and BV605-conjugated CD68 (1:400, BioLegend, San Diego, CA, USA) antibodies for 20 min at 4 °C in the dark. At least 10,000 events were recorded from each hemispheric sample using an LSR Fortessa flow cytometer (BD Biosciences, San Jose, CA, USA) and analyzed with FlowJo software v10.4. Please see Appendix A for detailed methods.

### 4.7. Immunofluorescent Staining

Mouse brains were fixed and collected after transcardial perfusions with ice-cold PBS and 4% paraformaldehyde (PFA), as described previously [13,26]. The brains were post-fixed overnight in 4% PFA and cryoprotected in 30% sucrose before being sectioned at 25 μm thickness using a Leica SM2010R microtome (Leica, Wetzlar, Germany) for immunofluorescent staining. Fluorescent images were captured under a 40× objective lens using a Nikon A1R confocal microscope (Nikon, Melville, NY, USA). Identical digital imaging acquisition parameters were used, and images were obtained and analyzed in a blind manner throughout the study. Please see Appendix A for detailed methods.

### 4.8. Data Analysis

An unbiased study design and analyses were used in all the experiments. The blinding of investigators to the experimental groups was maintained until the data were fully analyzed whenever possible. Data were expressed as the mean ± SD or SEM, and all data were tested for a normal distribution. Not normally distributed data were analyzed with a Mann–Whitney U test or other appropriate alternative tests according to the data (GraphPad Prism, La Jolla, CA, USA). A two-tailed Student’s *t*-test with 95% confidence was used when comparing two conditions. For more than two conditions, a one-way or two-way ANOVA analysis was used, depending on the data. A *p* value < 0.05 was considered statistically significant. All data were included unless the appropriate outlier analysis suggested otherwise.

## 5. Conclusions

Recanalization interventions with short therapeutic windows are the only treatments available for ischemic stroke patients. Targeting inflammation presents a promising treatment for stroke patients with a wider therapeutic window; however, no therapy is currently available for reducing neuroinflammation after stroke. In the current study, the blockade of NHE1 protein activity with HOE642 (Cariporide) or Rimeporide at 24 h post-stroke displayed no acute neuroprotection but reduced inflammatory responses, enhanced white matter repair, and improved motor and cognitive function recovery up to 1 month after stroke. Our study identifies targeting NHE1 protein as a novel strategy to reduce neuroinflammation and white matter damage to improve post-stroke recovery.

## Figures and Tables

**Figure 1 ijms-24-13289-f001:**
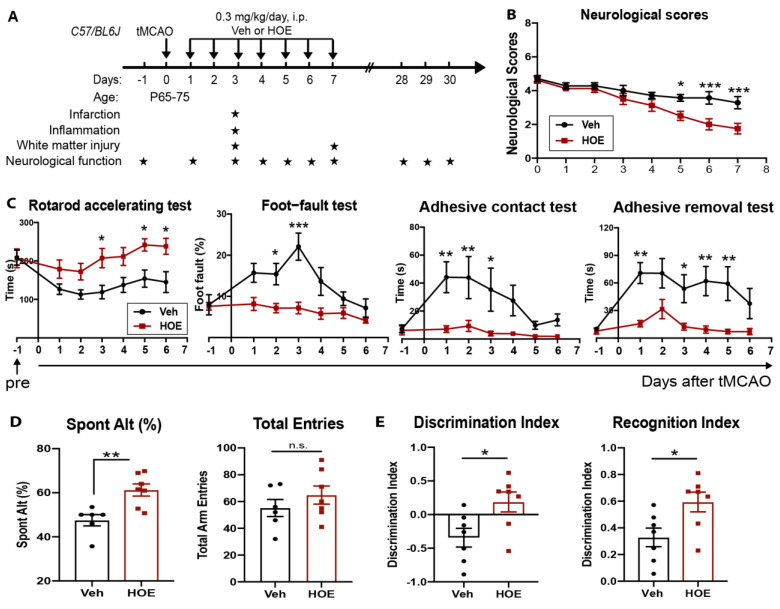
Administration of NHE1 inhibitor HOE642 at 24 h post-stroke improved motor-sensory and cognitive functions in C57/BL6J mice. (**A**) Experimental protocol. (**B**) Neurological scores of mice at 1–7 d post-stroke. (**C**) Rotarod accelerating test, foot-fault test, adhesive contact test, and adhesive removal test results of the same cohort of mice in (**B**). (**D**,**E**) Y-maze spontaneous alternation test and novel object recognition test results at 28–30 d post-stroke in the same cohort of mice in (**B**). N = 7–8. Data are the mean ± SEM. * *p* < 0.05, ** *p* < 0.01, *** *p* < 0.001, n.s. non-significant.

**Figure 2 ijms-24-13289-f002:**
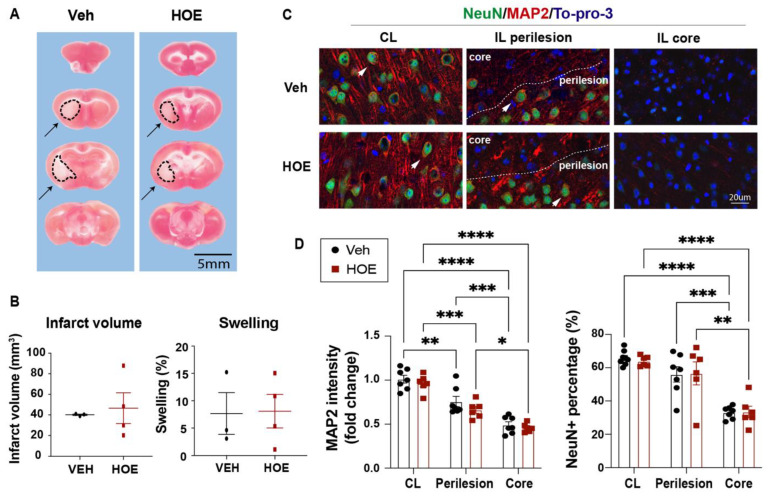
No changes in ischemic infarct or neuronal death in C57/BL6J mice treated with HOE642. (**A**) Representative images of TTC-stained brain sections of the Veh- and HOE-treated mice at 3 d post-stroke. Dotted line and arrows: infarcted tissues. (**B**) Infarct volume and brain swelling analysis of the TTC-stained brain sections. N = 3–4. Data are the mean ± SD. (**C**) Representative images of MAP2 and NeuN immunostaining in the contralateral (CL), ipsilateral (IL) perilesion area and ischemic core of the Veh- and HOE-treated mice at 3 d post-stroke. Arrowheads: preserved NeuN^+^ neurons. (**D**) Quantitative analysis of MAP2 intensity or NeuN^+^ cell counts in the Veh- and HOE-treated mice at 3 d post-stroke. N = 6–7. Data are the mean ± SEM. * *p* < 0.05, ** *p* < 0.01, *** *p* < 0.001, **** *p* < 0.0001.

**Figure 3 ijms-24-13289-f003:**
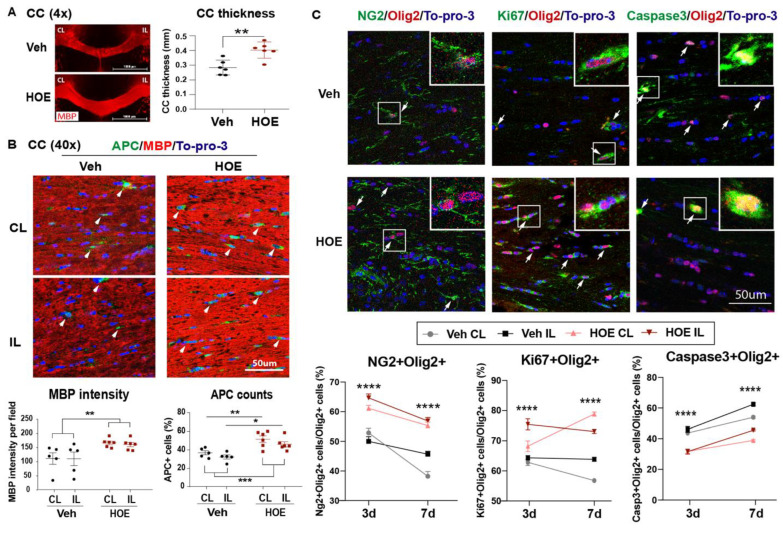
Post-stroke administration of HOE642 enhanced white matter myelination and oligodendrogenesis. (**A**) Representative corpus collosum (CC) images stained with MBP (4×) and quantitative analysis of CC thickness in Veh- or HOE-treated brains at 3 d post-stroke. Scale bar: 1000 μm. (**B**) Representative confocal CC images (40×) and quantitative analysis of MBP intensity and APC^+^ counts in Veh- and HOE-treated brains at 3 d post-stroke. Arrowheads: APC^+^ cells. N = 5–6. (**C**) Representative images and quantitative analysis of NG2^+^Olig2^+^, Ki67^+^Olig2^+^, and Caspase3^+^Olig2^+^ cells in the CL and IL hemispheres of the CC at 3 d post-stroke. Arrows: double positive cells. N = 6–7. Data are the mean ± SEM. * *p* < 0.05, ** *p* < 0.01, *** *p* < 0.001, **** *p* < 0.0001. Individual data points for 3 d and 7 d post-stroke are shown in Appendix A.

**Figure 4 ijms-24-13289-f004:**
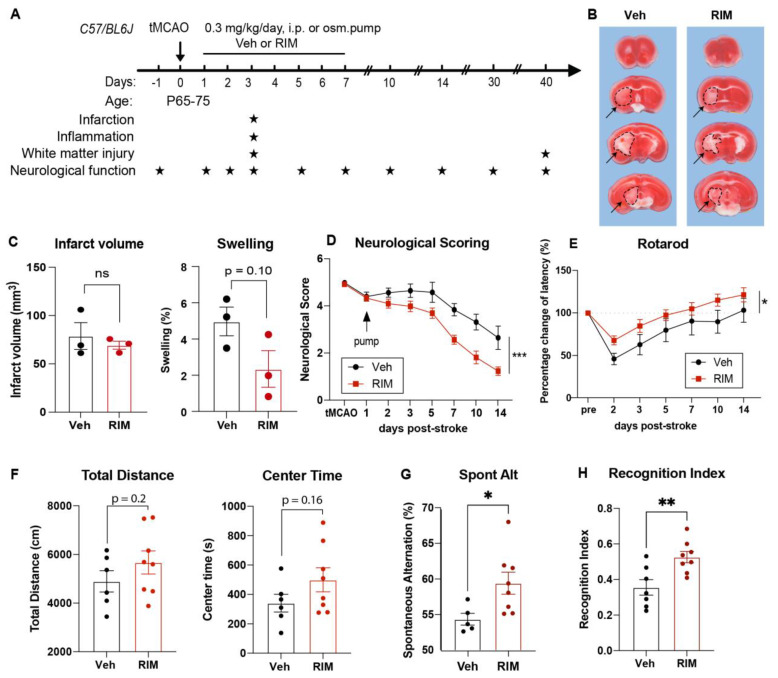
Efficacy of the post-stroke delivery of the novel NHE1 inhibitor Rimeporide. (**A**) Experimental protocol. (**B**,**C**) TTC staining and infarct volume and brain swelling analysis of the Veh- and RIM-treated mice at 3 d post-stroke. N = 3. Dotted line and arrow: infarct tissues. (**D**,**E**) Neurological scoring and rotarod accelerating test results for motor-sensory functions in Veh- or RIM-treated mice from 1 to 14 d post-stroke. N = 9 for Veh, N = 10 for RIM. (**F**–**H**) Open field test, Y-maze spontaneous alternation test (at 28-30 d post-stroke), and novel spatial recognition test results (at 37-40 d post-stroke) in the same cohort of mice in (**D**,**E**) Data are the mean ± SEM. * *p* < 0.05, ** *p* < 0.01, *** *p* < 0.001.

**Figure 5 ijms-24-13289-f005:**
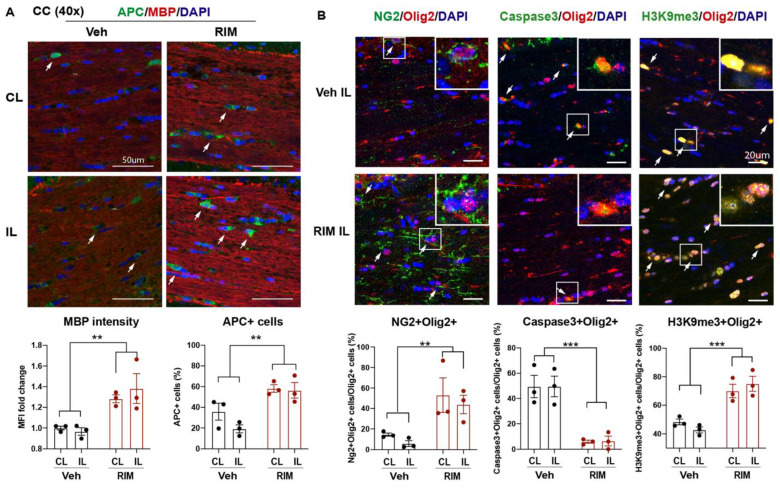
Post-stroke delivery of Rimeporide improved white matter myelination after stroke. (**A**) Representative images and quantitative analysis of MBP intensity and APC^+^ counts in the Veh- and RIM-treated brains at 3 d post-stroke. Arrows: APC+ cells. N = 3. (**B**) Representative images and quantitative analysis of NG2^+^Olig2^+^, Caspase3^+^Olig2^+^, and H3K9me3^+^Olig2^+^ cells in the CL and IL hemispheres of the CC at 3 d post-stroke. Arrows: double positive cells. N = 3. Data are the mean ± SEM. ** *p* < 0.01, *** *p* < 0.001.

**Figure 6 ijms-24-13289-f006:**
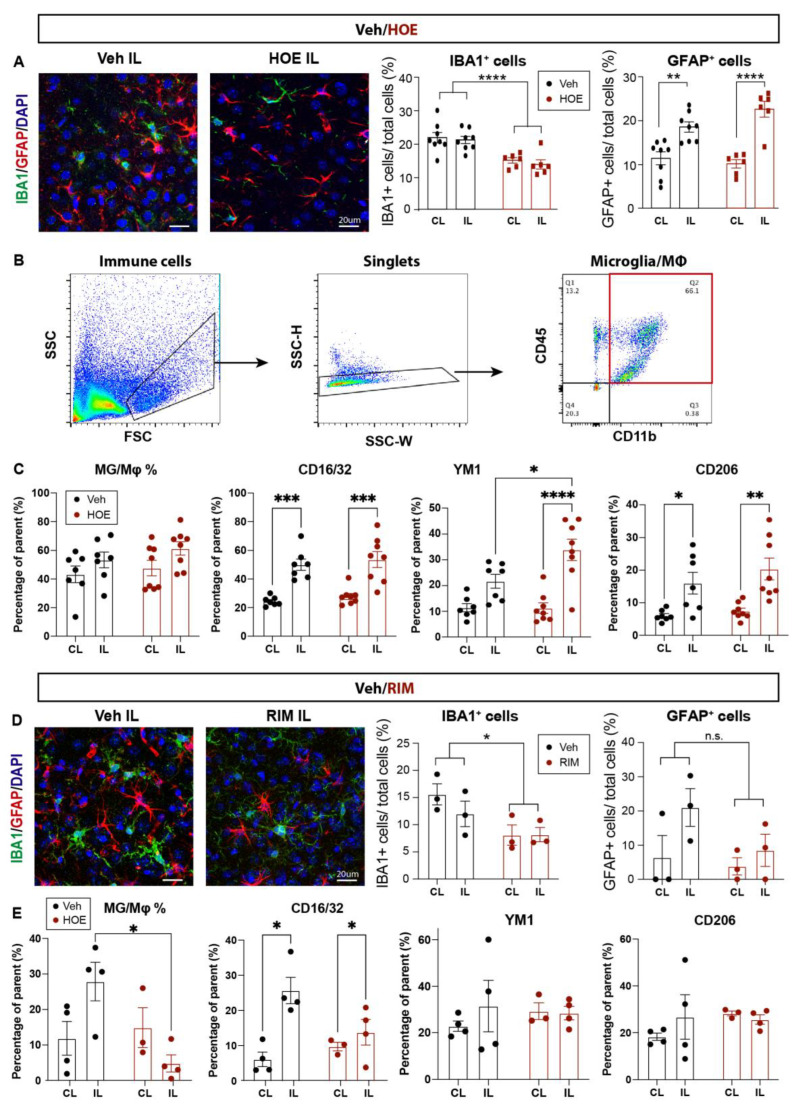
Post-stroke administration of HOE642 or Rimeporide reduced microglial inflammation without affecting astrogliosis (**A**). Representative images and quantitative analysis of GFAP^+^ and IBA1^+^ cells in the CL and IL cortex of Veh- and HOE-treated mice at 3 d post-stroke. (**B**) Representative gating strategy for CD11b^+^CD45^+^ microglia/macrophages. (**C**) Quantitative analysis of inflammatory profiling markers within parent CD11b^+^CD45^+^ microglia/macrophages in Veh- and HOE-treated mice at 3 d post-stroke. (**D**) Representative images and quantitative analysis of GFAP^+^ and IBA1^+^ cells in the CL and IL cortex of Veh- and RIM-treated mice at 3 d post-stroke. (**E**) Quantitative analysis of inflammatory profiling markers within parent CD11b^+^CD45^+^ microglia/macrophages of Veh- and RIM-treated mice at 3 d post-stroke. * *p* < 0.05, ** *p* < 0.01, *** *p* < 0.001, **** *p* < 0.0001.

## Data Availability

Data is contained within the article or Appendix A.

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
