# Peer review of "Pharmacological Inhibition of NHE1 Protein Increases White Matter Resilience and Neurofunctional Recovery after Ischemic Stroke"

_ijms, 2023, doi:10.3390/ijms241713289_

Round 1
Reviewer 1 Report
This is a comprehensive study on the effects of NHE1 inhibitors, HOE642 and Rimerporide on motor and cognitive recovery, inflammatory microglial activation and white matter damage in murine model for ischemic stroke.
Overall, the study was carefully designed and the abundance of interesting results is a testament of the efforts and care the authors have put into the study, and for that the authors are to be commended.
Just a few questions/issues:
1) The order for the various sections in the Materials and Methods is confusing. For example, the section "Transient focal schema" should be before drug administration while neurological function tests should be before 2,3,5-triphenyltetrazolium chloride (TTC) staining.
2) Why were these particular concentrations of HOE642 and Rimerporide chosen for the study?
3) It will be helpful to the readers if the authors can explain what the various scores and index measure for the various neurological function test such as the adhesive contact test, adhesive removal test and Y-maze spontaneous alternation test for Results section 3.1
4) Can the authors explain why the results for the neurological tests were more detailed for HOE642 than Rimerporide e.g. for the HOE642 study, there are results for foot-fault test, adhesive contact test and adhesive contact test (Figure 1C) were not shown for Rimerporide (Figure 1E). Similarly, what were the reasons HOE642 mice and Rimerporide mice subjected to different test (Figure 1D and Figure 3F). Is there any difference between the effects of HOE642 and Rimerporide and if so would it be beneficial to combine these two drugs?
Author Response
1) The order for the various sections in the Materials and Methods is confusing. For example, the section "Transient focal schema" should be before drug administration while neurological function tests should be before 2,3,5-triphenyltetrazolium chloride (TTC) staining.
Response: We thank the reviewer to point this out. As suggested, we have adjusted the order in Materials and Methods.
2) Why were these particular concentrations of HOE642 and Rimerporide chosen for the study?
Response: We thank the reviewer for this important question. We have added the following information regarding the selected drug concentrations in line 84-87: “…as we recently reported that HOE642 administration (at the 0.3 mg/kg dose) reduced microglial activation [28] and was neuroprotective in a traumatic brain injury (TBI) mouse model [29].” and in line 166-168 “Osmotic mini-pumps with Veh (5% DMSO in PBS) or RIM at 2.1 mg/kg (equivalent to 0.3 mg/kg/day) were implanted subcutaneously in mice at 1-7 d after stroke (Figure. 4A).” Efficacy of three different doses of RIM (0.3, 0.5, and 1 mg/kg/day) in tMCAO mice was tested in pilot experiments (data not shown) and 0.3 mg/kg/day of RIM was chosen accordingly, which has been included on line 279-280: “We chose a low dose regimen of 0.3 mg/kg/day after pilot experiments with 0.3, 0.5 and 1 mg/kg/day (data not shown).”
3) It will be helpful to the readers if the authors can explain what the various scores and index measure for the various neurological function test such as the adhesive contact test, adhesive removal test and Y-maze spontaneous alternation test for Results section 3.1
Response: As suggested by the reviewer, we included short explanations for the various behavioral tests in the Result section 2.1, line 89-99: “Neurological scoring is a well-established neurological deficit grading system in evaluating neurological deficits after tMCAO, ranging from 0 for no neurological deficit to 8 where stroke-related death occurs [13, 28).” and “These HOE642-treated mice also exhibited significant improvements in motor-sensory functions with longer latency of staying on the rotating rods in the rotarod accelerating test as early as 3 d post-stroke (p < 0.05), with less missed steps in foot-fault test at 2 d post-stroke (p < 0.05), and with increased sensitivity to the sticker in adhesive contact test and faster removal of the adhesive tape in adhesive removal test at 1 d post-stroke (p < 0.01) (Figure. 1C).”
4) Can the authors explain why the results for the neurological tests were more detailed for HOE642 than Rimerporide e.g. for the HOE642 study, there are results for foot-fault test, adhesive contact test and adhesive contact test (Figure 1C) were not shown for Rimerporide (Figure 1E). Similarly, what were the reasons HOE642 mice and Rimerporide mice subjected to different test (Figure 1D and Figure 3F). Is there any difference between the effects of HOE642 and Rimerporide and if so would it be beneficial to combine these two drugs?
Response: We thank the reviewer to point out this overlook. We have included the results for foot fault test and adhesive contact/removal tests for Rimeporide in Supplemental Figure 2B. We also revised the paragraph describing the neurological tests for RIM-treated mice in line 171-178: “In addition, the RIM-treated mice exhibited significantly faster motor function improvements in neurological scoring and rotarod accelerating test from 2-14 d post-stroke (Figure. 5D-E), but not the foot-fault test or adhesive tests (Figure. S2B). These RIM-treated mice also showed significantly improved memory function in the y-maze spontaneous alternation test and the novel spatial recognition test at 30 d post-stroke, compared to the Veh-treated mice (p < 0.05, Figure. 4H-I), and slight improvement in locomotor activity and reduced anxiety (although did not reach a statistical significance, Figure. 5F).”
Reviewer 2 Report
This is a well-written manuscript. The experimental designs and data interpretation are generally sound. However, there are a few concerns that need to be addressed:
1. The format: the format of the manuscript does not follow the journal guideline. The result section should immediately follow the introduction.
2. The sample image used in Figure 2 and Figure 4 represented two extremes of the transient MCAO in C57Blk mice. Figure 2 showed no cortical infarct and only damages to caudates. Figure 4 showed extensive cortical infarct. Although it is understood that there are unavoidable variabilities with tMCAO model in mice, picking the two extreme cases as representative images is not the best option. These images should be replaced with more representative ones.
3. The tune regarding NHE1 inhibitors as a potential stroke treatment may be too optimistic. Cariporide is an old drug that has failed clinical trials. The potential caveats for NHE1 inhibitors as stroke treatments should also be discussed to bring in a little more balance.
Author Response
1) The format: the format of the manuscript does not follow the journal guideline. The result section should immediately follow the introduction.
Response: We thank the reviewer to point this out. As suggested, we have adjusted the order of the sections.
2) The sample image used in Figure 2 and Figure 4 represented two extremes of the transient MCAO in C57Blk mice. Figure 2 showed no cortical infarct and only damages to caudates. Figure 4 showed extensive cortical infarct. Although it is understood that there are unavoidable variabilities with tMCAO model in mice, picking the two extreme cases as representative images is not the best option. These images should be replaced with more representative ones.
Response: We thank the reviewer for this suggestion. We have replaced the representative TTC images in Figure 4.
3) The tune regarding NHE1 inhibitors as a potential stroke treatment may be too optimistic. Cariporide is an old drug that has failed clinical trials. The potential caveats for NHE1 inhibitors as stroke treatments should also be discussed to bring in a little more balance.
Response: We have included discussion on the clinical trials that Cariporide has failed in, in line 261-274: “HOE642 has been tested in the GUARDIAN (Guard During Ischemia Against Necrosis) phase II/III clinical trial for myocardial infarction, showing a 25% reduction of risk in the high-risk patients and reported no serious adverse events in the cariporide-treated group (with a dose of 120 mg, intravenous t.i.d. for 2-7 days) over placebo [35]. However, despite of reduced myocardial infarction, it failed in the EXPEDITION (Sodium-Proton Exchange Inhibition to Prevent Coronary Events in Acute Cardiac Conditions) phase III clinical trial due to increased embolic stroke incidents [36], which is likely due to the excessive high dose (180 mg loading dose, then 40 mg/h in 24 h, and 20 mg/h in 48 h, intravenous) and its paradoxical effects on platelet activation [36, 37]. Our study with a low dose of HOE642 administered at 24 h post-stroke indicated the therapeutic potential of pharmacological blockade of NHE1 protein with minimized potential adverse effects. Additional studies are warranted on whether the delayed administration regimen with low doses of NHE1 inhibitors has any adverse effects on platelet or other coagulation dysregulation.”
Potential caveats for systemic administration of NHE1 inhibitors after stroke have also been revised and discussed in line 336-349: “The weakness of this study lies in the systemic NHE1 protein blockade with HOE642 or Rimeporide administration. OPCs/OLs have comparably high levels of expressions of Nhe1 gene [51], which is a dominant regulator for their intracellular pH (pHi) [52, 53]. Systemic inhibition of NHE1 protein could directly affect OPCs/OLs and exert impact on remyelination. We observed the apparent improvements in white matter repair with increased CC thickness, MBP protein expression and mature OL preservation in the NHE1 inhibitors-treated groups. However, whether pathological upregulation of NHE1 protein occurs in OPCs/OLs after stroke and whether HOE642 or RIM directly inhibits NHE1 protein in these cells to promote remyelination warrants further investigation in future studies. In light of the phenotypes of global transgenic knockout of Nhe1 on increased excitability of the CNS and epilepsy (especially slow wave epilepsy) in mice [54], it raises additional concerns about the global administration approach of NHE1 inhibitors. Development of more cell-type specific targeted approach and specific post-stroke treatment regimen is warranted to minimize these adverse effects.”